# Organic photovoltaic mini-module providing more than 5000 V for energy autonomy of dielectric elastomer actuators

Ershuai Jiang [1,2,6], Armin Jamali [1,3,6], Mathias List[2,4], Dushyant Bhagwan Mishra[1,3], Seyed Alireza Sheikholeslami[1,3], Frank Goldschmidtboeing [1,3], Peter Woias[1,3], Clemens Baretzky[2,4], Oliver Fischer [2,5], Birger Zimmermann[2], Stefan W. Glunz [1,2,5] & Uli Würfel [1,2,4] ✉

Dielectric elastomer actuators (DEAs) are widely used for soft robotics. The required voltages of over 1000 V are usually supplied by amplifiers with batteries or power grids which however have limited operation time or mobility. This problem also exists for other advanced mobile devices such as electro-aerodynamic thrusters. This work reports on the development of high-voltage organic photovoltaic mini-modules (HV-OPMs) comprising 5024 individual sub-cells on an area of $3.8 \times 3.9$ cm$^2$. Under 100 klux white LED illumination, an open-circuit voltage ($V_{OC}$) of 5534 V and an efficiency of 6.4% is achieved with the photoactive material PM6:GS-ISO whereas with PV-X plus a $V_{OC}$ of 3970 V and an efficiency of 19.0% is obtained. Furthermore, a soft suction cup based on DEA was built and could successfully be powered with one of these modules. These results show that HV-OPMs are very promising to realize energy autonomy of low-power high-voltage devices.

Electric energy is the most convenient power source in modern civilization. For different devices, the demands regarding voltage, power, and portability (mobility) are quite different. Generally, low-power low-voltage portable devices are usually powered by batteries, while unportable/immobile devices/equipment are powered by an electrical (power) grid. In these conventional scenarios, the high-voltage devices/equipment are usually high-power too. However, there are more and more high-voltage but low-power devices developed in recent years, such as e.g., dielectric elastomer actuators (DEAs)[1–3], electro-aerodynamic propulsion (EAD thruster)[4,5], piezocomposite ornithopter, etc. The functions of these devices require them to be highly mobile, but this is confined by the conventional power supply, which boosts voltage via voltage amplifiers with either an electrical grid or

batteries[6,7]. For many devices, both batteries and voltage amplifiers are bulky. To maintain their functionality, the devices have to be tethered with long wires or untethered but powered by tiny batteries with the drawback of rather small capacitance. For example, Chen et al. realized a controllable flight of a microrobot based on DEA artificial muscles. However, the working voltage above 1000 V was supplied by wires that were twisted quickly after takeoff, making any long-distance flight rather impossible[8]. Ji et al. developed a fast, untethered insect soft robot with DEA. Since the on-board battery is small, the continuous working time is limited to only 14 min[9]. Thus, there is a trade-off between mobility and working duration. To overcome this problem, it is timely to develop a durable and portable low-power high-voltage source that can be operated under typical ambient conditions.

[1]Cluster of Excellence livMatS, University of Freiburg, Georges-Köhler-Allee 105, Freiburg, Germany. [2]Fraunhofer Institute for Solar Energy Systems ISE, Heidenhofstr. 2, Freiburg, Germany. [3]Faculty of Engineering, Department of Microsystems Engineering (IMTEK), University of Freiburg, Georges-Köhler-Allee 101, Freiburg, Germany. [4]Freiburg Materials Research Center FMF, University of Freiburg, Stefan-Meier-Str. 21, Freiburg, Germany. [5]Chair of Photovoltaic Energy Conversion, Department of Sustainable Systems Engineering INATECH, University of Freiburg, Emmy-Noether-Str. 2, Freiburg, Germany. [6]These authors contributed equally: Ershuai Jiang, Armin Jamali. ✉e-mail: uli.wuerfel@ise.fraunhofer.de

A creative idea came from Wang's group, which developed triboelectric nanogenerators to power DEAs[10,11]. However, this type of power supply is based on motion (vibration) and needs high vibration amplitudes and large areas to provide relevant power. Harvesting light from the environment (either indoor or outdoor) by photovoltaics is instead a common way to realize energy autonomy. As long as illumination is available, solar cells can provide continuous energy output in contrast to energy supplies based on storage, such as batteries or supercapacitors.

A single-junction solar cell generates a voltage in the order of 1 V. Although a multi-junction solar cell can achieve an open-circuit voltage ($V_{OC}$) of several volts, this is still far too low for the high-voltage applications mentioned above. In order to increase the voltage output, a solar module is required which is made by interconnecting individual solar cells laterally in series, thus adding up the voltage. In the following, we provide an overview of published work on small-sized solar modules with high voltages. As early as 1995, Lee et al. developed a miniaturized high-voltage solar cell array with triple-junction amorphous silicon sub-cells[12]. The $V_{OC}$ of this solar cell array was 150 V, which can be used to power microelectromechanical systems (MEMS). Similarly, Nam et al. fabricated an a-Si solar module with a $V_{OC}$ of about 100 V on a 2.5 cm$^2$ substrate[13]. Furthermore, Hung et al. fabricated a mini c-Si solar module on a chip with a reported value of 3.13 V/mm$^2$ for the specific $V_{OC}$[14]. However, one issue for silicon-based solar mini-modules is that the edge effect has a detrimental impact on the performance, and the specific edge increases dramatically with decreasing active area for each sub-cell[15]. Furthermore, most silicon-based mini-modules were built with silicon on insulator (SOI) wafers, which require a complex fabrication process[16]. On the other hand, organic photovoltaics (OPV) offer the benefit of easy coating processes and the possibility of fabricating mechanically flexible devices. Lim et al. demonstrated a mini-module with 300 sub-cells with a spin-coated organic absorber material on an active area of 45 mm$^2$, providing a $V_{OC}$ of 90 V[17]. Furthermore, Niggemann et al. even achieved a $V_{OC}$ of 880 V with nano structure[18]. These results show that OPVs are well suited to realize high voltage on small areas. However, most of the high-voltage mini-modules mentioned above require a complex photolithography process to pattern precise microstructures. Another common problem is that the power conversion efficiency ($PCE$) of these OPV devices is rather low (generally lower than 1%). Very recently, a $V_{OC}$ over 1000 V with a high $PCE$ of over 19% has been achieved by our group with OPV mini-modules comprising 1640 interconnected small cells realized by efficient laser patterning[19]. This constitutes a major step toward developing sustainable, low-power high-voltage sources.

As for many applications, a voltage of 1000 V is still not sufficient, this work reports on the development and application of high-voltage organic photovoltaic mini-modules (HV-OPMs) with even higher voltage. The HV-OPMs were fabricated by laser structuring, resulting in 5024 sub-cells interconnected in series on an area of 3.8 × 3.9 cm$^2$. Under 100 klux warm white LED illumination, the champion device based on the organic absorber material PV-X plus (details on the materials is provided in the Methods Section) gives a $V_{OC}$ of 3970 V with a $PCE$ of 19.0%, while the PM6:GS-ISO based module has a $V_{OC}$ of 5534 V (specific $V_{OC}$ of 373 V/cm$^2$) with a $PCE$ of 6.4%. These are the highest voltages achieved to date with photovoltaic modules. In addition to characterizing the HV-OPM with current-voltage ($IV$) curves under different illumination intensities, electroluminescence (EL) and dark lock-in thermography (DLIT) imaging were performed to analyze in detail the voltage losses occurring in the mini-modules. Furthermore, high reliabilities could be shown in experiments on partial shading, potential induced degradation, breakdown voltage, and stability. As a use case, a DEA suction cup was built and powered by one HV-OPM based on PM6:GS-ISO, and the energy flow was characterized in detail. Finally, the gripping of a 28 g object by the DEA suction cup powered by the HV-OPM was successfully demonstrated. It is worth noting that the HV-OPMs could also be used as a solid-state high-voltage source for many other applications, such as e.g., EAD thrusters, piezocomposite ornithopters, etc.

## Results and discussion

### High-voltage organic photovoltaic mini-module design and fabrication

In previous reports, the silicon based HV-OPMs fabrication was enabled by peel-off patterning technique, which is time consuming and complex. Moreover, this method is generally not suitable for OPVs, since the solvent used for photoresist could damage organic layers. Although multilevel peel-off patterning was already successfully developed for OPVs, the complex process makes it still not be widely adapted[20]. On the contrary, direct laser patterning is powerful for micron-scale structuring, which is widely used for thin film based high efficiency PV module fabrications, due to its low cost, fast processing and minimized geometrical loss[21–23]. Details on fabricating HV-OPMs can be found in the Method section. Figure 1a shows the photograph of a PM6:GS-ISO based HV-OPM with its 5024 small solar cells distributed over 32 columns and 157 rows, all interconnected in series. A microscopic image of the mini-module is presented in Fig. 1b, where the alignment of the different laser patterns (P1–P4) can be seen. For adjacent two sub-cells within a row, the bottom ITO and top Au/Ag electrodes were separated by P1 and P3 lines, respectively, and interconnected by a P2 line. Two adjacent rows were separated by a P4 line. For such mini-modules, the geometric fill factor ($GFF$), which is the ratio of active area to total area (shown exemplarily for one sub-cell in Fig. 1c), has to be maximized. For this design, the dead area in the lateral direction ($W_{dead}$) includes P1, P3 and the area between them which is 0.06 mm wide. In the vertical direction, the dead area ($H_{dead}$) is the P4 pattern, being 0.02 mm wide. The $GFF$ is related to both the sub-cell density ($N$, in cm$^{-2}$) and the dimensions of the sub-cell:

$$GFF = (W - W_{dead})\left(\frac{1}{NW} - H_{dead}\right)N \qquad (1)$$

As shown in Fig. 1d, the $GFF$ mapping is plotted as a function of both $W$ and $N$, with the maximum $GFF$ ($GFF_{max}$) for each $N$ plotted as the yellow line. In our case, $N \approx 333$ cm$^{-2}$, $W = 1.2$ mm and H = 0.25 mm, resulting in a $GFF$ of 0.874, close to the $GFF_{max}$ line. This is a remarkable value for a small area device with more than 5000 individual cells. Very recently, an improved laser processing has been shown to enable ablation holes smaller than 100 nm, which could possibly further reduce the dead area and thus increase the $GFF$ (Supplementary Fig. 1)[24].

The fabrication process is illustrated in Fig. 1e. The P1 lines were structured on a full-area indium tin oxide (ITO) substrate with a width of about 10 μm. The functional layers of electron transport layer (ETL), photoactive layer (PAL) and hole transport layer (HTL) were spin-coated subsequently. It is well-known that shunting is a common issue for indoor OPV devices as the impact of the shunt resistance becomes dominant at low light intensity[25]. To reduce the effect of shunts under low illuminance, zinc oxide (ZnO) was chosen as the ETL because of its superior shunt-proof property (Supplementary Fig. 2)[26]. For the PV-X plus device, an additional organic ETL, namely PFN-Br was coated after the ZnO to improve the contact with the photoactive layer[27]. However, PFN-Br is sensitive to the high temperature (160 °C) that was applied for annealing PM6:GS-ISO (Supplementary Fig. 3). Therefore, for the PM6:GS-ISO devices, only ZnO was used as the ETL. The alignment of P1, P2, and P3 is the same as for common thin film solar modules, but the perpendicular P4 lines were grooved through all the layers from the top electrode down to the ITO, defining 157 rows in our case. In each of these rows, 32 sub-cells are interconnected in series, and the

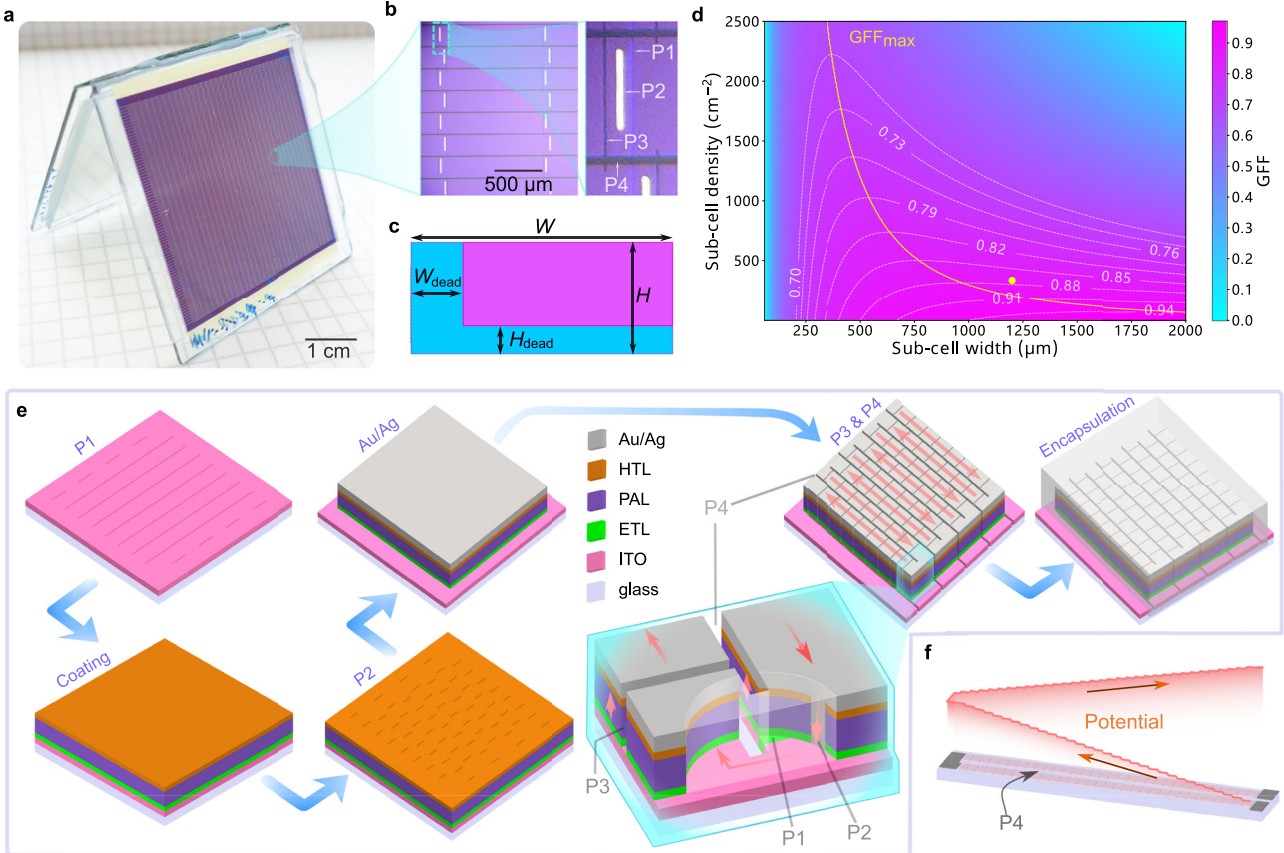

**Fig. 1 | Structure and fabrication process of the HV-OPMs. a** Photograph and (**b**) microscopic image of an HV-OPM based on PM6:GS-ISO with the laser patterns (P1-4). **c** Schematic diagram of the dead area (blue) and active area (purple) in one sub-cell (not drawn to scale), where the $W_{dead}$ represents the width of the dead area between laser patterns of P1 and P3, while $H_{dead}$ is the P4. **d** Calculated geometric fill factor (*GFF*) as a function of sub-cell density (*N*) and sub-cell width (*W*), using fixed

$W_{dead} = 60\,\mu m$ and $H_{dead} = 20\,\mu m$. The yellow dot shows the actual *GFF* of the HV-OPMs in this study. **e** Schematic diagrams of the fabrication process and interconnection of two sub-cells at the end of two rows with alignment of P1–P4. **f** Schematic image of the potential distribution (when illuminated) along two rows which are separated on the right (P4 extends until the edge) and connected at the left (P4 line ends before, see also (**e**)).

different rows are interconnected in a zigzag pattern. The current flow direction is illustrated by red arrows in the cross section sub-plot that shows the alignment of P1–P4 clearly. This is achieved by the P4 line which extends to the edge of the substrate on one side whereas it ends at the interconnection pad on the other side, see Fig. 1f. For this reason, the potential difference across a P4 line equals the voltage of one sub-cell on one side, while it equals the voltage of 64 sub-cells at the other. Under 100 klux, the sum of the $V_{OC}$ of 64 sub-cells is approximately 75.7 V for PM6:GS-ISO and 53.2 V for PV-X plus devices. This means that the electric field might be stronger than the electric breakdown threshold of air (about 30 kV/cm), leading to a risk of spark discharge between two rows. This can, however, be avoided by filling the P4 lines with a material that has a higher electric breakdown strength. For this purpose, an epoxy resin based encapsulant (LP655) was used. The breakdown strength of the LP655 was measured to be about 600 kV/cm (Supplementary Fig. 4). This means that the width of the P4 lines could even be further reduced to decrease the dead area and thus increase the *GFF* and sub-cell density without risking discharge sparks.

**Photovoltaic performance**

To evaluate the performance of HV-OPMs, *IV* measurements were performed with a warm white LED lamp. The irradiance spectrum of the LED lamp is shown in Fig. 2a, with commonly used AM1.5 G for comparison. Figure 2b shows the external quantum efficiency (EQE) of the solar cells based on PV-X plus and PM6:-GS-ISO. *IV* curves of the HV-OPMs were measured in a wide range of illuminances from 1 klux to 100 klux, as shown in Fig. 2c, d. From the *IV* curves of the HV-OPMs, the

photovoltaic parameters $I_{SC}$, $V_{OC}$, *FF*, $P_{max}$ and *PCE* were extracted (Fig. 2e). The detailed values of all modules are listed in Supplementary Table 1. As the laser processing can lead to efficiency losses, we intended to estimate the latter in a quantitative manner. For this, we fabricated small single solar cells with identical dimensions as the sub-cells in the HV-OPMs (Supplementary Fig. 5 and Supplementary Table 2). From the *IV* curves of these single solar cells we extrapolated data for the HV-OPMs and plotted them as dashed lines in Fig. 2.

Figure 2e top panel shows $V_{OC}$ as a function of the illuminance. Both PV-X plus and PM6:GS-ISO based HV-OPMs show a slightly lower $V_{OC}$ compared to the extrapolated values, namely (relative) 4.9% and 6.9% less at 100 klux for the best devices, respectively. The origin of these losses will be discussed in the next section. Furthermore, illuminance dependent $I_{SC}$ curves are also shown. Since $I_{SC}$ is linearly dependent on the illuminance (as well as $P_{max}$), double logarithmic sub-plots are shown. At 100 lux, the average $I_{SC}$ for PV-X based HV-OPMs is 28.9 μA. For PV-X plus there is a perfect agreement while the measured $I_{SC}$ values of PM6:GS-ISO HV-OPMs are significantly lower than the extrapolated values. It is important to note that the single cells were not subjected to the P2 patterning step and therefore the pho-toactive layer was not exposed to air before the deposition of the top metal electrode. In contrast, in the fabrication process of the modules, the P2 patterning step is performed (before deposition of the top metal electrode) in ambient air. Therefore, ozone molecules can penetrate the photoactive layer. Van der Pol et al. have reported that the higher density of states (DOS) caused by the reaction of PM6 with ozone can increase recombination and hinder charge transport,

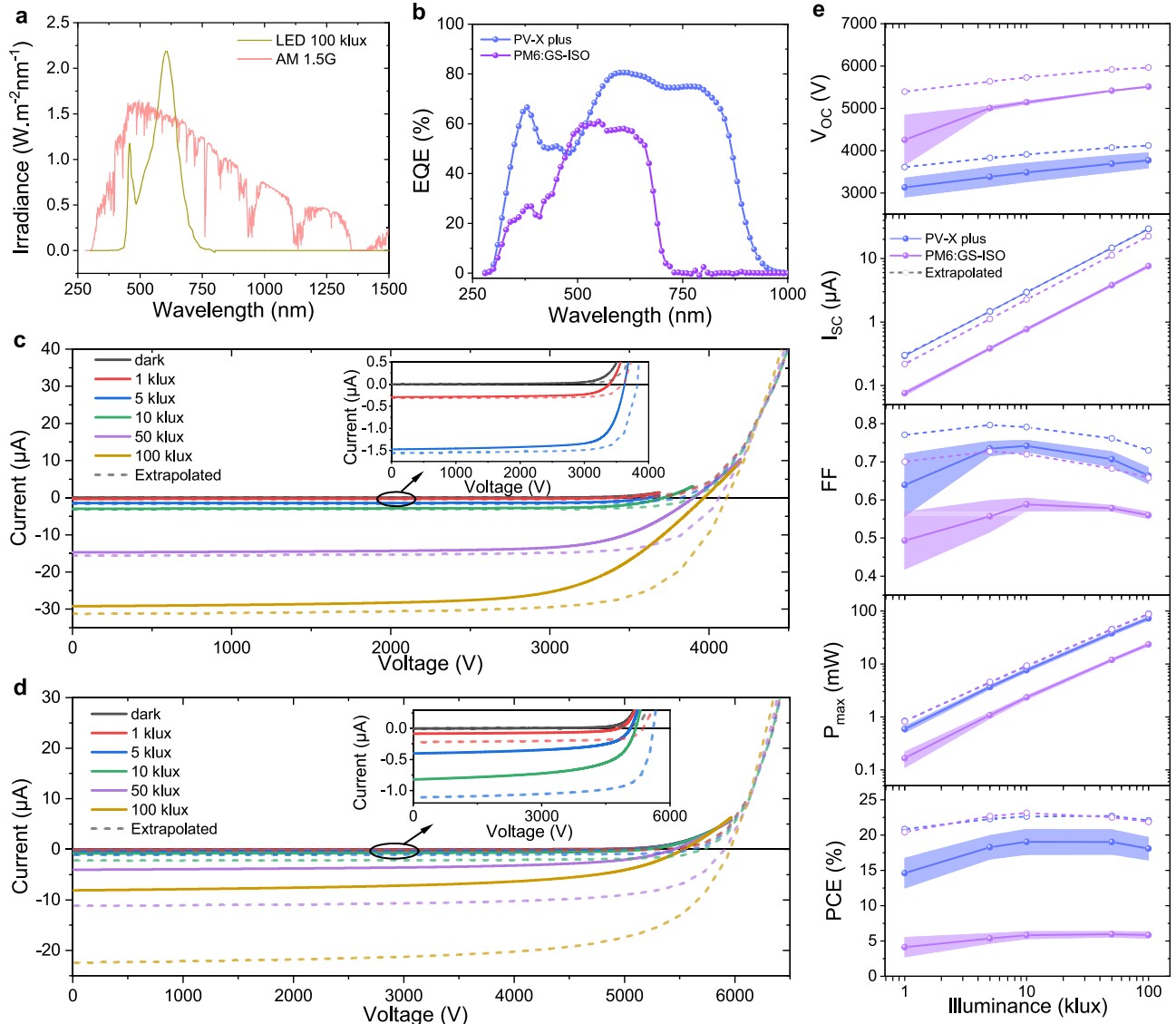

**Fig. 2 | Photovoltaic performance of the HV-OPMs. a** Irradiance spectra of LED lamp at 100 klux and AM1.5 G. **b** EQE spectra of PV-X plus and PM6:GS-ISO based solar cells. **c**, **d** *IV* curves of champion devices based on PV-X plus and PM6:GS-ISO, respectively. **e** Statistical results of the open-circuit voltage ($V_{OC}$), short-circuit current ($I_{SC}$), fill factor ($FF$), maximum power ($P_{max}$), and power conversion efficiency ($PCE$) of the HV-OPMs, the solid lines with spheres are the average values of three samples, while shaded areas indicate standard deviations. For (**c**–**e**), the dashed lines are values extrapolated from single cells (see text).

making it the main cause for the reduction of OPV performance in air[28]. Our experiments also support this idea (Supplementary Figs. 6, 7). It can be expected that the problem will be solved by laser processing of P2-P4 under inert atmosphere. This is however beyond the scope of this study. The *FF* of all the HV-OPMs exhibit a concave downward trend in the Fig. 2e. This can be consistently explained by the fact that under low illumination, the *FF* is dominated by shunts, while under high illumination, it is limited by the series resistance ($R_s$) which for OPVs is dominated by the photoactive layer[29–31]. Therefore, the *PCE* (according to active area, determined by multiplying the active area of a sub-cell with 5024, resulting 13.17 cm²) of the HV-OPMs shows a similar trend, as shown in the bottom panel in the Fig. 2e. Nevertheless, the *PCE* of the PV-X plus HV-OPMs is high, even above 20% (champion device) in the illuminance range from 5 klux to 50 klux. At 100 klux, although the *FF* is only 0.657, the *PCE* is still 19.0%, with a $V_{OC}$ of 3970 V, $I_{SC}$ of 29.2 μA and $P_{max}$ of 76.2 mW. This $P_{max}$ is sufficient for the energy supply requirements of many centimeter-scale devices, making such a

module a promising candidate for replacing batteries, high-voltage converters, or high voltage supply wires[9,32–34]. It is worth noting that the high *PCE* originates from strongly reduced thermalization losses due to the much narrower spectrum of the LED illumination compared to AM1.5 G. The average *PCE* of PV-X plus based HV-OPM under AM1.5 G would be 10.2% (see Supplementary Note 1 and Supplementary Fig. 8 for calculation). Under 1 klux indoor illumination, the module gives a *PCE* of 17.6%, with a $V_{OC}$ of 3387 V, $I_{SC}$ of 0.293 μA, *FF* of 0.710 and $P_{max}$ of 0.70 mW. Thus, even at this low illuminance the $P_{max}$ is still sufficient to power applications that consume power in the μW range[35,36]. As mentioned above, the PM6:GS-ISO HV-OPMs suffer significantly from air exposure during the P2 process in air, which leads to much lower *PCE*, and thus $P_{max}$. Nevertheless, it still shows a reasonable *PCE* of 6.4% at 100 klux, with a $V_{OC}$ of 5534 V, $I_{SC}$ of 8.1 μA, *FF* of 0.570 and $P_{max}$ of 25.6 mW. This makes it possible to power small size ionic thrusters or solid-state fans based on the electrohydrodynamic effect[37–39].

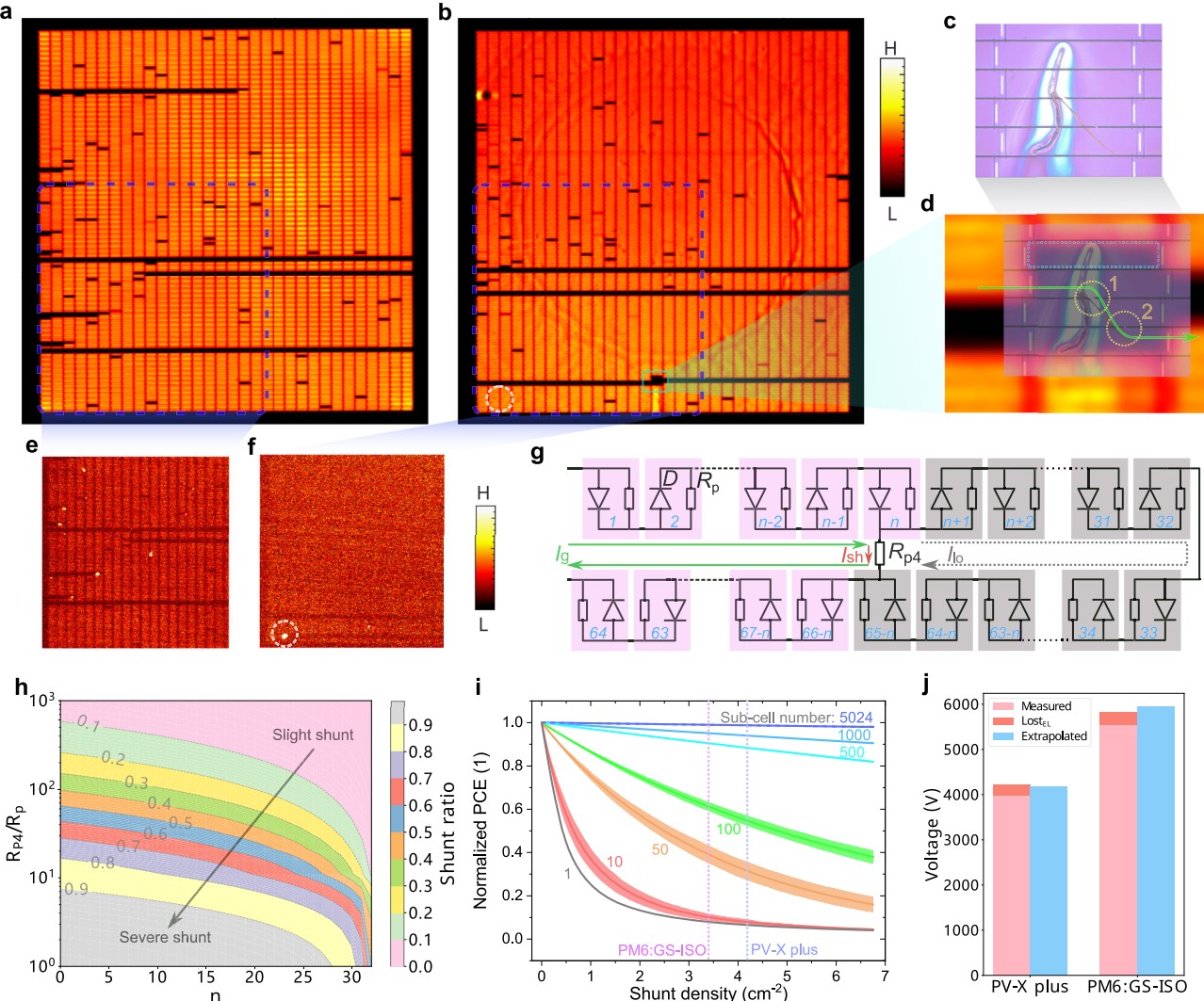

**Fig. 3 | Shunt analysis for the HV-OPMs. a**, **b** Electroluminescence (EL) images of the PV-X plus and PM6:GS-ISO based HV-OPMs, respectively. **c** Microscopic image of a fiber dust, localized in the green frame in (**b**). **d** Zoomed-in image of the green framed area in (**b**) overlapped with the image from (**c**) with the green arrow showing the direction of current flow and the yellow circles indicating the shunts. **e**, **f** Dark lock-in thermography (DLIT) images corresponding to the blue framed area in (**a**) and (**b**). **g** Simplified equivalent circuit of two adjacent rows of sub-cells in the HV-OPM when performing EL imaging with a shunt ($R_{P4}$) between sub-cells number $n$ and $65\text{-}n$, the green arrow showing global current ($I_g$), the red arrow illustrating shunt current ($I_{sh}$) and the dashed gray arrow presenting the local current through the dark sub-cells ($I_{lo}$). **h** Calculated shunt ratio as a function of shunt position and $R_{P4}/R_p$ ratio. **i** Normalized power conversion efficiency (*PCE*) as a function of shunt density for a module with different numbers of sub-cells, with the vertical dot lines are determined by the numbers of the DSC$_{single}$ in (**a**, **b**). **j**, Histogram of $V_{OC}$ losses, showing the measured $V_{OC}$ under 100 klux, $V_{OC}$ losses determined by EL imaging (Lost$_{EL}$), and $V_{OC}$ extrapolated from single solar cells.

## Shunt analysis by electroluminescence imaging

Exploring the performance losses induced by the processing is essential for the further improvement of the HV-OPMs, but it is rarely reported. In fact, there are no studies in literature describing a detailed characterization of high-voltage mini-modules. Here, we employed electroluminescence (EL) and dark lock-in thermography (DLIT) imaging to detect improperly working sub-cells within the HV-OPMs. Figure 3a, b show the EL images of a PV-X plus and a PM6:GS-ISO based HV-OPMs, respectively, where most sub-cells are bright, i.e., they function normally. However, there are also some dark sub-cells, which means that they hardly contribute to the voltage output of the modules. The total number of dark sub-cells is 293 for PV-X plus and 240 for PM6:GS-ISO based HV-OPM, thus about 5% of the total sub-cell number (5024). Generally, the dark sub-cells (DSCs) can be divided into two types, named DSC$_{single}$ and DSC$_{inter-row}$ here. DSC$_{single}$ refers to single sub-cells individually distributed while DSC$_{inter-row}$ refers to neighbored sub-cells of two adjacent rows. Analysis of the data shows that

DSC$_{inter-row}$ makes up for roughly 80% of all dark sub-cells for both types of HV-OPMs. Figure 3c shows a microscopic image of a dust fiber localized in the green framed area. To analyze the defect more clearly, we made it semitransparent and stacked it with the zoomed-in EL image of the corresponding area, see Fig. 3d. In contrast to typical DSC$_{inter-row}$ in two adjacent rows, three rows are affected here with two clear shunts due to the dust fiber. More examples of shunts can be seen in Supplementary Fig. 9.

Figure 3e, f show DLIT images corresponding to the blue framed areas in Fig. 3a, b, where the normal working sub-cells appear orange, while the non-functioning sub-cells are gray or dark, with bright spots as shunts. In Fig. 3e, it is straightforward to correlate the sub-cells to the corresponding EL image. However, Fig. 3f is much more blur which makes it harder. Although, there is a clear DSC$_{inter-row}$ shunt marked by the white circle but cannot be seen at the corresponding position in the EL image (white circle in Fig. 3b). Since the DLIT imaging was done after the EL imaging and as furthermore a much higher voltage was

applied (7000 V), it is reasonable to assume that the shunt was generated in the very process of DLIT imaging. This means that the P4 lines are weak points and shunting may occur during operation, even with encapsulation. This problem may in future be resolved by widening the P4 lines.

To develop a deeper understanding of the impact of the $DSC_{inter-row}$ during EL and DLIT imaging, a simplified equivalent circuit of two adjacent rows of the HV-OPMs is shown in Fig. 3g, in which a sub-cell is represented by a diode ($D$) and a parallel resistor ($R_p$). A resistor ($R_{P4}$) between the sub-cells $n$ and 65-$n$ is used to represent the resistance of the shunt in the P4 line. Accordingly, the global current ($I_g$) can be divided into the shunt current ($I_{sh}$) through $R_{P4}$ and the local current ($I_{lo}$) flowing through the dark sub-cells. Then, the shunt ratio ($r_{sh} = I_{sh}/I_g$) is calculated as a function of shunt position ($n$) and the ratio of $R_{P4}/R_p$:

$$r_{sh} = \frac{65 - 2n}{65 - 2n + \frac{R_{P4}}{R_p}} \qquad (2)$$

As shown in Fig. 3h, to maintain a low $r_{sh}$ (< 0.1), of the ratio $R_{P4}/R_p$ has to be as high as about 600, depending on the position of the shunt.

The $DSC_{single}$ is typically surrounded by bright sub-cells. This originates from the local nature of most defects. As a matter of fact, the area affected by such a shunt can be much larger than the actual defect itself due to the high conductivity of the electrodes. However, the sub-cells in our HV-OPM have only a very small area that thus strongly restricts the detrimental effect of a local shunt. To quantify this correlation, we simulated the normalized *PCE* of modules as a function of sub-cell numbers and shunt density ($D_{sh}$) on an area of 3.8 × 3.9 cm². As shown in Fig. 3i, for a module with 10 sub-cells, the normalized *PCE* strongly depends on the $D_{sh}$. This is one of the main reasons that ITO-based solar mini-modules often have low performance under low illumination intensity[40]. For a fixed $D_{sh}$ around 4 cm⁻² (close to real conditions), we obviously find that the larger the number of sub-cells, the higher the (normalized) *PCE*. This clearly shows why the HV-OPMs can maintain a remarkably high efficiency despite the occurrence of local shunts.

With $N_d$ being the number of dark sub-cells and assuming further that they do not generate any voltage, the total voltage loss is given by:

$$\Delta V_{OC,loss} = N_d V_{OC,1} \qquad (3)$$

where $V_{OC,1}$ is the open-circuit voltage of a single cell. Next, we compared the measured $V_{OC}$, $\Delta V_{OC,loss}$, and the extrapolated $V_{OC}$ for an illuminance of 100 klux. As presented in Fig. 3j, the sum of measured $V_{OC}$ and $\Delta V_{OC,loss}$ is quite close to the extrapolated $V_{OC}$ for the PV-X plus based mini-module. For PM6:GS-ISO, the difference is slightly larger. We attribute this to the higher sensitivity of PM6:GS-ISO to air exposure as explained in the previous section.

## Reliability analysis

For high-voltage solar modules or arrays, potential induced degradation (PID) is a common issue, which has been studied deeply for c-Si, CIGS and perovskite solar cells (PSCs), but not yet for OPVs[41–43]. To investigate it, we carried out PID tests as it is commonly done, i.e., by application of strong external electric fields perpendicular to the surface of single solar cells (Fig. 4a)[42,44,45]. The results show that there is no obvious PID being observed for both photoactive materials (Fig. 4b). Strong electric fields also exist around P4 lines (especially at the separation end) under operating conditions. Thus, we fabricated 2-row mini-modules with 128 sub-cells and aged them under 50 klux LED illumination and both-open circuit and short-circuit conditions. As shown in Fig. 4c–f, the PM6:GS-ISO modules show quite stable $V_{OC}$ output after a slight burn-in degradation in the beginning, but the PV-X plus modules show significant degradation, without a clear difference

between open-circuit and short-circuit conditions. A similar trend was also observed for $I_{SC}$, *FF*, and *PCE*. It is worth noting that in the first several hours, the *FF* and, thus *PCE* increased compared to their initial values. We attribute this to the shunt-burning effect during the measurements. EL imaging also indicates that the severely degraded PV-X plus sub-cells are more likely distributed in one row (Supplementary Fig. 10), but a more detailed analysis is beyond the scope of this work.

Breakdown under reverse bias of sub-cells caused by partial shading is another major risk for real applications of solar modules (Fig. 4g). Firstly, we measured reverse breakdown voltages ($V_{br}$) and power densities ($P_{br}$) of single solar cells. Average values were determined to 41.7 V and 4.8 W/cm² for PV-X plus and 36.2 V and 1.1 W/cm² for PM6:GS-ISO, respectively (Supplementary Fig. 11 and Supplementary Table 3). To the best of our knowledge, the measured $V_{br}$ are much higher than those of c-Si solar cells (usually about 15 V) and PSCs (generally lower than 10 V)[44–46]. Then, we calculated a breakdown index ($BRI$) as a function of shaded sub-cell number ($N_{sha}$) under 100 klux and AM1.5G as:

$$BRI = \frac{I_{sha}V_{sha} - N_{sha}A_{sub-cell}P_{br}}{N_{sha}A_{sub-cell}P_{br}} \qquad (4)$$

where $I_{sha}$ and $V_{sha}$ are current and voltage applied to the shaded sub-cells and $A_{sub-cell}$ the area of sub-cells (Supplementary Note 2 and Supplementary Figs. 12–15). As shown in Fig. 4h, the $BRI$ has a negative value for both PM6:GS-ISO and PV-X plus HV-OPMs for any value of $N_{sha}$, which means that there is no breakdown risk at all for illuminances of either AM1.5 G or 100 klux LED.

## Powering a DEA suction cup with a HV-OPM

Up to date, few efforts have been made to apply solar cells for high-voltage devices, including DEAs and ultralight micro aerial vehicle[47,48]. However, in these reports, the high-voltage was still generated by high-voltage convertors. To demonstrate a meaningful use case for the organic photovoltaic mini-modules developed in this work, a DEA-based soft suction cup for the handling of fragile objects was fabricated, using a multilayer technique (Supplementary Figs. 16, 17)[49]. Figure 5a shows the structure of the DEA suction cup, while the working principles are presented in Supplementary Note 3 and Supplementary Fig. 18. To demonstrate how the DEA suction cup can generate a suction pressure powered only by one HV-OPM without any need for batteries and electronics, we first characterized the suction pressure of the suction cup under different illuminations. As presented in Fig. 5b, the maximum suction pressure was 476 Pa under 100 klux. With decreasing illuminance, the suction pressure was reduced and thus a longer time was required to reach equilibrium (Supplementary Fig. 19). Then, the suction cup was mounted on a vertical positioning stage to demonstrate the object lifting (Fig. 5c). Below the suction cup, an object (20 g weight in an 8 g holder) was set on an electrical weight scale. From left to right, Fig. 5d shows the demonstration steps (see also Supplementary Video). It can be seen that the object was successfully lifted up after the light has been switched on.

To explore more dynamic details of the demonstration setup, we monitored the transient currents and voltages of the HV-OPM and DEA suction cup after turning on the light with an oscilloscope. The corresponding equivalent circuit is shown in Fig. 5e. Therein, the voltage detection was realized by implementing a resistor ($R_L$) of 96.6 GΩ connected in series with one channel ($Ch_1$) of the oscilloscope, while the current through the DEA is detected through another channel ($Ch_2$). The HV-OPM, consisting of serially interconnected solar sub-cells, was electrically modeled by a total series resistance $R_{s,OPM}$, a total parallel resistance $R_{p,OPM}$, a total diode $D_{OPM}$, and a photocurrent source $I_{ph}$ (Supplementary Note 4, Supplementary Fig. 20 and Supplementary Table 4).

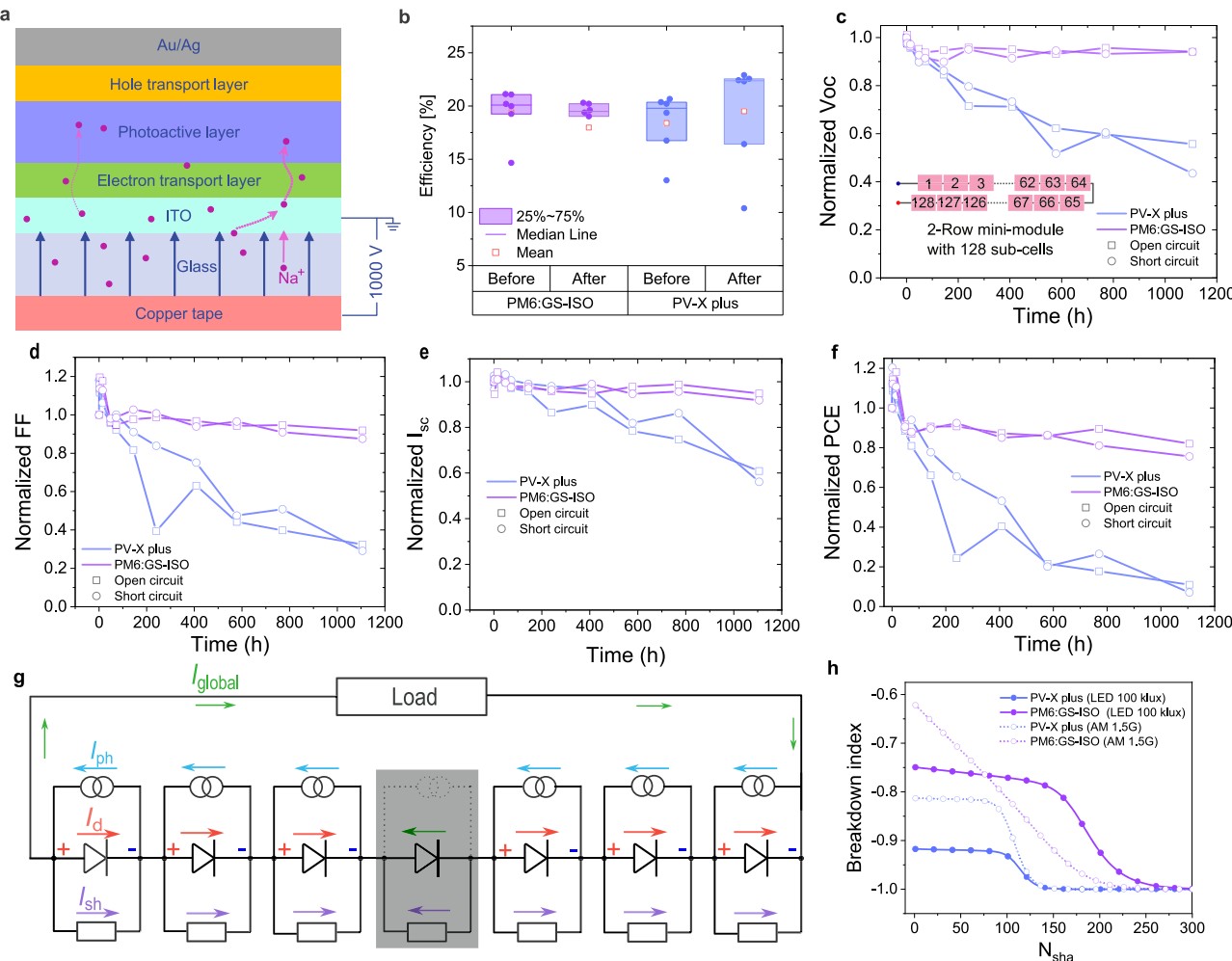

**Fig. 4 | Reliability analysis. a** Schematic image of the potential induced degradation (PID) test. **b** Efficiency of the solar cells before and after the PID test. **c**–**f** Normalized $V_{OC}$, $FF$, $I_{SC}$, and $PCE$ stability of 2-row mini-modules with 128 sub-cells under 50 klux LED illumination aged either under open- or short-circuit conditions, respectively. **g** Equivalent circuit of a module with one shaded sub-cell being reverse biased by non-shaded sub-cells. **h** Breakdown index as a function of the number of shaded sub-cells ($N_{sha}$).

The equivalent circuit of the DEA suction cup includes a series resistance $R_{s,DEA}$, a capacitance $C_{DEA}$ and a parallel resistance ($R_{p,DEA}$). The $R_{s,DEA}$, and $R_{p,DEA}$ are constant, while the $C_{DEA}$ is a function of applied voltage $V_{DEA}$:

$$\frac{C_{DEA,0}}{C_{DEA}}\left(1 - \sqrt{\frac{C_{DEA,0}}{C_{DEA}}}\right) = \frac{1}{2}\frac{\varepsilon_r \varepsilon_0}{Y z_0^2} V_{DEA}^2 \qquad (5)$$

where $C_{DEA,0}$ is the initial capacitance, $\varepsilon_r \varepsilon_0$ the dielectric constant, $Y$ the elastic modulus and $z_0$ the initial electrode distance of the DEA suction cup (Supplementary Note 5 and Supplementary Table 5). The electrical modeling is completed by adding the Kirchhoff voltage equation:

$$V_{HV-OPM} = R_{s,DEA} I_{DEA} + V_{DEA} \qquad (6)$$

and the Kirchhoff current equation:

$$I_{HV-OPM} = C_{DEA}\dot{V}_{DEA} + \dot{C}_{DEA}V_{DEA} + \frac{V_{DEA}}{R_{p,DEA}} + \frac{V_{HV-OPM}}{R_L} \qquad (7)$$

at the connection node. Here, the effect of the oscilloscope inputs is omitted since it is very small. The model was derived for the case of an unloaded (i.e., non-gripping) suction cup. This means that the DEA is virtually free to move, and no external force by the produced suction pressure has to be considered (Supplementary Note 6 and Supplementary Fig. 21). Figure 5f shows the simulated and measured charging curves of the non-gripping DEA. For high illuminances of 50 klux and 100 klux, charging occurs in less than 1 s and a maximum $V_{DEA}$ of more than 5 kV is reached. At 1 klux, it took up to 30 s to fully charge the DEA due to the photogenerated current being much smaller.

In Fig. 5g, the measured and simulated charging powers under different illuminances are plotted. It can be seen that the peak values are reproduced quite accurately, and they are very close to the $P_{max}$ of the HV-OPM as extracted from $IV$ curves (Supplementary Fig. 22). This means that the peak power transfer efficiency from the HV-OPM to DEA is close to 100%. However, after the peaks, certain offsets are visible. Figure 5h shows the current, voltage and power transients of the HV-OPM in the demonstration system after turning on the lamp (100 klux) together with the suction pressure transient. It can be noted that there is a fast charging phase until the voltage reaches its plateau after about 0.25 s. After that, a second slower phase starts which leads to further changes in suction pressure. We attribute this second phase to the viscoelastic behavior of the DEA (Supplementary Note 7 and Supplementary Figs. 23, 24). As this was not regarded in the

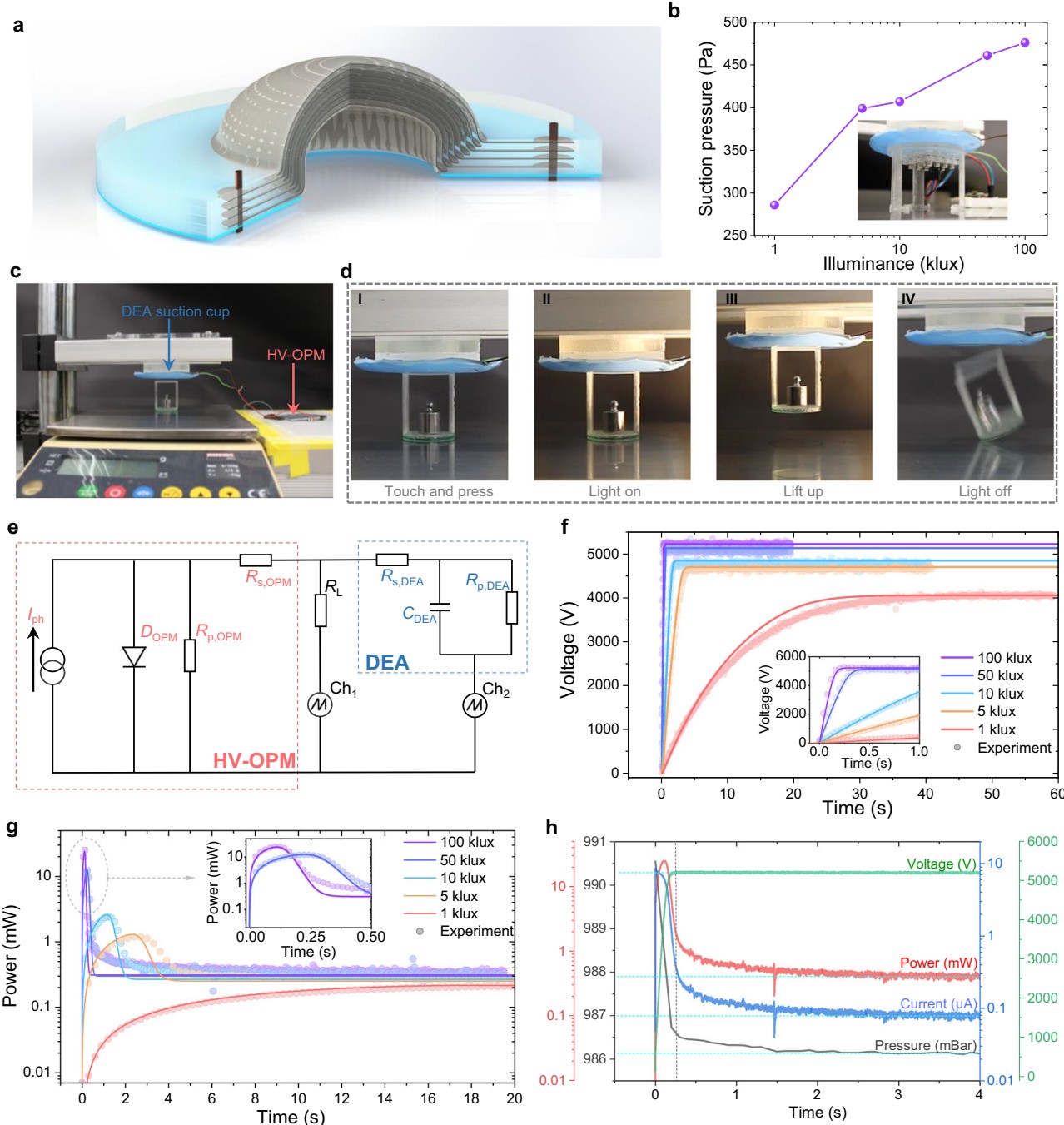

**Fig. 5 | Characterization of the DEA suction cup powered by an HV-OPM.**
**a** Schematic structure of the DEA suction cup. **b** Suction pressure as a function of illuminance, with the inset showing the measurement setup. **c** Setup used for lifting an object by the HV-OPM powered DEA suction cup. **d** Demonstration steps I to IV. **e** Equivalent circuit of the HV-OPM and DEA suction cup system, including the measurement inputs. **f** Simulated (solid lines) and measured (symbols) voltage transients of the DEA suction cup after turning on the lamp with different illuminances. Electrical power of the HV-OPM under different illuminances. **h** Suction pressure of the DEA suction cup, current, voltage, and power changing of the HV-OPM after turning on the lamp (100 klux).

simulations, it is the reason for the offsets between simulation and experiment in Fig. 5g.

Notably, the current in Fig. 5h shows an unexpected overshoot (better visible in Supplementary Fig. 20), which could be proven to originate from an overshoot of the illumination intensity when turning on the lamp (Supplementary Fig. 25).

In this work, high-voltage organic photovoltaic mini-modules are presented that can be used to realize energy autonomy of low-power high-voltage devices. Two organic absorber materials were used,

namely PV-X plus and PM6:GS-ISO, which can generate record-high voltages of 3970 V and 5534 V under 100 klux warm white LED light on an area of 3.8 × 3.9 cm², respectively. The PV-X plus based device shows a remarkable efficiency of 19%, while the PM6:GS-ISO based module achieves 6.4%. Advanced characterization based on dark lock-in thermography and electroluminescence imaging enabled a thorough analysis of the shunt induced voltage losses. As a demonstrator and potential use case, a suction cup based on a dielectric elastomer actuator was built and a setup for a lift test was developed. Directly

powered by a high-voltage organic photovoltaic mini-module, the DEA suction cup grasped and lifted a weight of 28 g successfully. The detailed power analysis shows that a single mini-module could in fact charge several DEAs in sequence and still provide enough power to compensate for resistive losses. Finally, a coupled simulation model for the combination of HV-OPM and DEA was developed that allows for the design of future generation actuation systems powered by HV-OPMs. To realize true applications of HV-OPMs, several challenges have to be overcome, such as processing on flexible or even ultra-flexible substrates, miniaturing control circuit, etc. These are however beyond the scope of this study and will be addressed in future work.

## Methods

### Materials

PV-X plus is a mixture of PV2300, PVA3 and N1100 that were purchased from Raynergy Tek. HTL-1 was purchased from Brilliant Matters Inc. and Zinc oxide (ZnO) nanoparticle dispersion/solution from Avantama. Poly[(9,9-bis(60-(N,N,N-trimethylammonium)hexyl)-2,7-fluorene)-alt-2,7-(9,9-dioctylfluorene)] (PFN-Br), poly[[4,8-bis[5-(2-ethylhexyl)-4-fluoro-2-thienyl]benzo[1,2-b:4,5-b′]dithiophene-2,6-diyl]-2,5-thiophenediyl [5,7-bis(2-ethylhexyl)-4,8-dioxo-4 H,8-H-benzo[1,2-c:4,5-c′]dithiophene-1,3-diyl]-2,5-thiophenediyl] (PM6) and GS-ISO were purchased from Solarmer Materials Inc. Anhydrous o-xylene was purchased from Sigma-Aldrich. Encapsulation glue LP655 was purchased from DELO Industrie Klebstoffe GmbH & Co. KGaA. Silicone elastomer (Ecoflex 00-10) and silicone thinner (SILICONE THINNER™) were ordered from KauPo Plankenhorn e.K.

### Organic photovoltaic mini-modules fabrication

The fabrication method of the photovoltaic mini-modules can be found in our previous work[19]. Generally, PV-X plus solution was prepared by mixing the components PV2300, PVA3 and N1100, with a ratio of 1:1:0.2 in o-xylene and a total concentration of 22 mg/ml. PFN-Br solution was prepared in anhydrous ethanol with a concentration of 0.5 mg/ml. PM6:GS-ISO solution was made by mixing PM6 and GS-ISO, with a ratio of 1:1.3 in o-xylene and a total concentration of 23 mg/ml. Indium tin oxide (ITO) glass substrates were cleaned in an ultrasonic bath subsequently in acetone, isopropanol and deionized water for 5 min each. This was followed by laser patterning (P1). The laser system used was a UV ultra-short pulse system from Lumemtum in conjunction with a fixed optics and an axis system from Aerotech "Pro Serie" to move the sample. For PV-X plus devices, ZnO/PFN-Br layers were spin coated as electron transport layer (ETL). For PM6:GS-ISO, only ZnO was coated as ETL. For absorber layer coating, PV-X plus solution was spin coated with a speed of 800 rpm, while PM6:GS-ISO was coated with 2500 rpm. The PM6:GS-ISO layer was annealed at 160 °C for 10 min. HTL-1 was spin coated as hole transport layer (HTL) for both type of devices, with a speed of 4000 rpm and annealed at 110 °C for 5 min. Next, all the spin coated functional layers were patterned (P2) by laser with a pitch of 3 μm and a pulse energy of 0.1 μJ. To form the top electrode, a 5 nm gold layer was thermally evaporated, followed by a 70 nm silver layer. Along with P1 and P2 patterns, P3 patterns were made to separate the top electrode, with a pulse energy of 0.082 μJ and a pitch of 3 μm. P4 patterns grooved all the layers from the top down to the glass substrates in the direction perpendicular to P1, P2 and P3, with a pulse energy of 0.25 μJ and a pitch of 2 μm. When structuring P3 and P4, the illumination side of the mini-modules was covered with a black tape in order to prevent an electrical breakdown. To encapsulate the mini-modules, LP655 glue was casted on the mini-modules directly, and a cover glass was used on top. Then the glue was pre-cured under the sun-simulator for 10 min and fully cured by a UV lamp for 5 min.

### Single organic solar cell fabrication

The small single solar cells were built on full area ITO glasses with a dimension of $25 \times 25 \, mm^2$. The coating and laser patterning

parameters were the same as those used for the HV-OPMs (for P1, P3 and P4). Since no interconnection is needed, there was no patterning of ETL/PAL/HTL layers, i.e., no P2 patterning step. The thermal evaporation of the Au/Ag layers (top electrode) was performed after the HTL coating without exposing the cells to air. The area of the single solar cell is 0.27 mm² for PV-X plus and 0.24 mm² for PM6:GS-ISO, according to the microscopic image (Supplementary Fig. 4). The single solar cells were encapsulated as the HV-OPMs

### Dielectric elastomer actuator suction cup fabrication

In this work, a silicone elastomer (Ecoflex 00-10) mixed with a silicone thinner were used and implemented the spin-coating technique to deposit thin layers with 550 μm thickness. We painted dry carbon black directly onto the elastomer layers to pattern ultra-thin electrode layers with a thickness of less than 5 μm. With this method, we first fabricated a flat planar actuator with 7 active layers of elastomer. To fabricate a fully soft suction cup from the planar actuator, we used a mold with a negative paraboloid cavity with an opening diameter of 22 mm and a height of 15 mm and applied an off-center layer of silicone as a backbone material to hold the structure of the suction cup.

### Current-voltage measurements

A 2700 K warm white LED lamp (LTAS-100/1 from Brandmaier) with a diffusion glass was used as light source. To determine the illuminance, we used a certified reference solar cell (RS-ID-4). To avoid the effect of environmental light, a black cloth was used to cover the samples when measuring. For the HV-OPMs, a two-channel source meter Keithley 2636 A was used. Channel A was used to provide input for a high voltage amplifier (10HVA24-P1) which can amplify the voltages up to a factor of 1000. The forward voltage scanning speed was 40 V/s, with a step size of 30 V and a dwell time of 0.15 s. The current was recorded with channel B during voltage sweeps. The measured data were calibrated by shifting the *IV* curves with an average current offset of 22.5 nA (see ref. 19. for details). The *IV* curves of the single solar cells were measured with a Keithley 2400 source meter, with a scanning speed of 0.67 V/s and a dwell time of 0.06 s. All the samples were tested in ambient air after encapsulation. With the measured *IV* curves, the *PCE* of the HV-OPMs under LED illumination were calculated as:

$$PCE = \frac{I_{mpp} \times V_{mpp}}{5024 \times A_{sub-cell} \times GFF \times P_{in}} \times 100\%$$

where $A_{sub-cell} = 1.2 \, mm \times 0.25 \, mm$, is the full area of a sub-cell, $P_{in}$ the light intensity under the respective illuminance (Supplementary Table 1), $I_{mpp}$ and $V_{mpp}$ are current and voltage at maximum power point, respectively.

### External quantum efficiency (EQE), Dark lock-in thermography (DLIT) and electroluminescence (EL) imaging

The EQE spectra were recorded at a LOANA solar cell analysis system from pv-tools. The bias voltages for DLIT and EL measurements were provided from the high voltage amplifier. DLIT measurements were conducted using a measurement setup from IRCAM GmbH. A Millenium 327k SM PRO IR Camera from IRCAM GmbH was used for image acquisition. The forward biases for DLIT measurements were 4000 V for PV-X plus and 7000 V for PM6:GS-ISO based modules, respectively. EL measurements were conducted on a measurement setup developed at Fraunhofer ISE and built by Intego GmbH. The images were acquired with a silicon charge-coupled device camera. The forward biases for EL measurements were 3500 V for PV-X plus and 5500 V for PM6:GS-ISO based modules, respectively.

### Simulation of the impact of localized shunts on the *PCE*

A computational model was used to simulate the distribution of shunts across various module configurations. Experimentally measured *IV*

data from the PM6:GS-ISO single solar cell served as the baseline input for a single diode model. A variable shunt was added to this model to generate *IV* curves with different shunt resistances. Each sub-cell within a module was then assigned an *IV* curve that included a shunt resistance based on the simulated distribution of shunts. Multiple module configurations, varying the number of sub-cells (10, 50, 100, 500, 1000, 5024), were evaluated, and 500 Monte Carlo simulations per configuration were conducted to account for the random distribution of shunts. The total voltage and current for the module were computed by aggregating the *IV* curves of all sub-cells, considering their respective shunt distributions. The module's power output was determined, and the normalized *PCE* was calculated by comparing it to the ideal, non-shunted scenario. For single-cell analysis, shunt resistances of 10 kΩ were considered, with varying shunt densities to observe their effects on *PCE*. The normalized *PCE* was plotted against the number of shunts per active area for all configurations, demonstrating the differences in performance across various configurations and highlighting the robustness of modules with a high number of individual cells against localized shunts.

### Potential induced degradation test

For potential induced degradation (PID) test, single solar cells with an area of 0.0925 cm$^2$ were fabricated (Supplementary Method). Before PID test, the solar cells were encapsulated. To build a strong electric field in the glass substrate and drive the Na$^+$ ions toward active layers, a copper tape was stacked on the surface of front (illumination) side, and 1000 V DC voltage between ITO layer and copper tape was supplied by a high voltage amplifier (10HVA24-P1). The PID test took in ambient air for three days (69 h). The *IV* measurement were done under an illumination of 1 klux.

### Irreversible breakdown test of solar cells

For breakdown under reverse bias test, single solar cells with an area of 0.0925 cm$^2$ were fabricated (Supplementary Method). The voltage sweeps from 1.5 V to − 45.0 V (for PV-X plus cells) or 35 V (for PM6:GS-ISO cells) with a step of 0.1 V and a source delay of 10 ms was generated by an SMU Keithley2400. The measurement took in dark.

### Stability under illumination test of 2-row mini-modules

2-row mini-modules with 128 sub-cells were fabricated with the procedure same for the HV-OPMs with 5024 sub-cells. The doubled subcell density makes 2-row mini-modules bearing a stronger electric field around the P4 line at the separation end, compared to the HV-OPMs. Then, the 2-row mini-modules were set on open-circuit or short-circuit and kept on illumination of 50 klux from a cool white LED lamp. The $V_{OC}$ were measured with the warm white LED lamp under 1 klux.

### DEA suction pressure measurement

To measure the suction pressure, we first connected one PM6:GS-ISO based HV-OPM directly to the suction cup. Afterward, we pressed and sealed the suction cup against a surface with an embedded pressure sensor. Then, the pressure under the dome was measured with a pressure sensor (MS5839-02BA, TE Connectivity) was installed inside. The pressure sensor was powered by an Arduino board connected to a computer. The pressure reading frequency was 10 Hz.

### Suction and lift-up demonstration

To demonstrate the suction and lift-up of objects, the DEA suction cup was mounted on an aluminum beam which can move up and down with a manually controlled screw bar. The object to be lifted includes two parts: 20 grams weight and 8 grams holder. Firstly, the suction cup was touched and pressed with the object. An initial load of about 1 N (i.e.,100 g) was applied to seal the suction cup. Then the suction cup was directly connected to a PM6:GS-ISO HV-OPM. After being illuminated under 100 klux LED light for 10 s, the suction cup and the gripped object were lift up slowly.

### Measurement of transient voltage and current of demonstration system

To measure the current through the DEA suction cup, an oscilloscope (PicoScope®S5444D) was connected in series with the DEA suction cup and the HV-OPM. The current was calculated from the recorded voltage using the input resistance ($R_{os} = 1$ MΩ). To measure the voltage of the DEA suction cup, a load resistance ($R_L = 96.6$ GΩ) was connected with another channel of the oscilloscope in series and in parallel with the HV-OPM. The voltage of the DEA was calculated by multiplying the recorded voltage with the ratio of $R_L/R_{os}$. The sample rate was 1 MS/s and quantization step size was 14-bit for all the measurements. The measurement range were 10 V for 100 klux, 5 V for 50 klux, 1 V for 10 klux, 0.5 V for 5 klux and 0.1 V for 1 klux when measuring the current through the DEA and 0.1 V for 100 klux-5 klux and 0.05 V for 1 klux when measuring the voltage of the DEA. The whole setup including the lamp, HV-OPM, load resistance and DEA suction cup were put in a Faradic cage to reduce noise.

### Transient voltage and current of demonstration system simulation

According to the circuit diagram in Fig. 5e, the Eqs. (5–7) build a firstorder nonlinear, temporal differential equation system for the four variables $V_1$(t), $V_{OPM}$ (t), $V_{DEA}$(t), and $I_{OPM}$(t), where $V_1$(t) is the voltage on the $R_{p,OPM}$. The parameters of the HV-OPM depend on the illuminance (Supplementary Table 4). This differential equation system was solved numerically in MATHEMATICA (Version 13.2, Wolfram Research) for chosen illuminances by the Newton method. The parameters for the DEA are given in Supplementary Table 5.

### Reporting summary

Further information on research design is available in the Nature Portfolio Reporting Summary linked to this article.

## Data availability

The authors declare all data supporting the findings of this study are available within the manuscript and Supplementary Information. Source data are provided in this paper under https://doi.org/10.6084/m9.figshare.26165398.

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

## Acknowledgements

This work was funded by the Deutsche Forschungsgemeinschaft (DFG, German Research Foundation) under Germany's Excellence Strategy—EXC-2193/1—390951807. The authors thank Andreas Nägele, René Haberstroh, and Lasse Bienkowski for laser patterning, Felix Martin for EQE measurement, and Lukas Klimmek for EL and DLIT measurements at Fraunhofer ISE.

## Author contributions

E.J., A.J., and U.W. conceived the study. E.J. and M.L. designed the HV-OPMs. E.J. fabricated the HV-OPVs and performed the related measurements. A.J., D.M., and S.S. fabricated the DEA suction cup and carried out the related characterizations. E.J. and A.J. did the demonstration of the HV-OPV powered DEA suction cup and performed the measurements. O.F. did the EL and DLIT measurements and contributed to their analysis. C.B. simulated the shunt density-dependent PCE. B.Z. contributed to the idea of investigating partial shading effects. F.G. simulated the transient voltage and power of the HV-OPM in the demonstration system. The manuscript was written by E.J., A.J., F.G., and U.W. The study was supervised by U.W., P.W., and S.G. All the authors contributed to the data analysis and discussion.

## Funding

## Competing interests

The authors declare that they have no competing interests.
