## [Transparent Peer Review file · Nature Communications]

Organic photovoltaic mini-module providing more than 5000 volts for energy autonomy of dielectric elastomer actuators

Corresponding Author: Dr Uli Würfel

Version 0:

Reviewer comments:

Reviewer #1

(Remarks to the Author)

This manuscript by Jiang et al. demonstrates a strategy for constructing ultra-high voltage OPV modules, with laser patterning as its core technique. While achieving such a high output voltage using OPV technology is indeed interesting, the manuscript appears to be heavily technical, lacking in scientific depth. Furthermore, concerns arise regarding the similarity of its core content to an ACS Energy Letters article previously published by the same research group (ACS Energy Lett. 9, 908–910 (2024)), with the primary distinction being an increased number of sub-cells per unit area, resulting in a higher open-circuit voltage. Despite more extensive characterization of the HV module and a proof-of-concept application, the innovative aspects of this work do not surpass those of the previous publication, casting doubt on its suitability for publication in Nature Communications. In my opinion, Scientific Reports may be a more appropriate journal for this research.

Recommendations for improvement:

1. It is recommended that the authors provide the EQE spectra of the devices, the LED lamp spectrum utilized for determining PCE, and a detailed explanation of the methodology employed for calculating PCE.
2. The authors should report the maximum output power of the module, as this is a crucial performance indicator.
3. While I acknowledge that shunting is less problematic in HV-devices with a large number of sub-cells, efforts to mitigate shunting and further enhance the performance of the HV-devices would strengthen the work.
4. Figures 2b and 2e should be revised for clarity. For instance, please ensure that the connection of adjacent positive and negative electrodes in the schematics is depicted more precisely and intuitively.
5. The authors have provided stability test results for HV-devices under 50k lux conditions. I suggest that, in addition to V_{oc} , data on the degradation of J_{sc} , FF, and PCE over time be included to provide a more comprehensive evaluation of device stability.

Reviewer #2

(Remarks to the Author)

The paper reports the fabrication and application of very high-voltage organic photovoltaic mini-modules under warm white LEDs. High voltages of 3970 V and 5534 V with two different active layers were achieved on about 15 cm² area. The characterization of dark lock-in thermography and electroluminescence imaging are clear for the understanding of shunt induced voltage losses. The demonstrator for driving a dielectric elastomer actuator is inspiring. I believe this is a piece of very quality work. The demonstration of the high voltage minimodules and its unique application are intriguing. I would recommend its publication in Nat. Communications. Minor comments:

Have the authors tried thicker active layers? Increasing the thickness of the active layers would reduce the possibility of the shunt devices.

Reviewer #3

(Remarks to the Author)

I read this paper with fun. This paper reports Organic photovoltaic mini-module providing more than 5000 volts. They

patterns P1-P4 lines using laser processes, and fabricated modules having impressively high Voc and PCE under 100 klux white LED. Finally, they show a demonstration of operating DEA actuator powered by only the OPV module under LED light illumination.

The concept is interesting, the fabrication to achieve high voltage using OPVs is well designed, analysis for the operation is well discussed. However, the reviewer suggests revisions before publication in Nature Communications.

1. The current description is reasonable if this paper is submitted to energy related journals since fabrication of high-voltage OPV module is challenging, but the motivation of such high-power output should be clarified more carefully to be published in Nature Communications. Although the authors mention "For many devices, both batteries and voltage amplifiers are bulky"; however, the OPV module used a glass substrate, which should be "bulky" also. Similarly, why do we need the DEA suction cup generating a suction pressure powered only by one HV-OPM without any need for batteries and electronic? What is the major difference between bulky batteries/electronic and rigid OPV module? Actually, there is a report of OPV-driven DEA actuator with support of circuits and batteries, being integrated to make a soft robot system (10.1038/s41598-024-60899-6). Is the procedure applicable to flexible substrates as the authors mentions "the possibility of fabricating mechanically flexible devices"? The relevant discussion should be added to make the justification for using these high voltage OPV modules clearer. Better way is to show the feasibility of these procedure using flexible substrates.

2. There are some recent papers related to OPV modules using laser patterning, which are recommended to be referred in this paper. Eg. doi: 10.1016/j.joule.2024.02.016, doi: org/10.1038/s41560-022-00997-9. The strength of laser patterning can be discussed with referring other patterning strategies such as doi: 10.1016/j.joule.2022.06.015.

3. Since Nature Communications have wider readership, detailed information should not be omitted. P1-P4 are understandable words for solar cell field researchers but are not at all for others. They used pico-second laser to make the patterns according to the reference 19, but this is not specified in this paper (Additionally, reference 19 does not show product details of the laser).

4. To me, 19% under 100 klux is too high since this is almost similar to 1-Sun condition, which maybe related to spectrum of LED light. The irradiance results of LED light with 100 klux (and others also) should be added to verify the calculation of efficiency. Additionally, I suggest showing the PCE under AM1.5G illumination conditions as well as LED light conditions to avoid misunderstanding.

5. "As shown in Fig. 5e, the BRI has a negative value for both PM6:GS-ISO and PV-X plus HV-OPMs for any value of Nsha, which means that there is no breakdown risk at all for illuminances of 100 klux and below" (Page 13); is there breakdown risk under AM1.5G condition?

6. Why is Fig.4f much more blur than Fig.4d? Is this related to the performance of the module?

7. The purpose of Figure 1 is not clear. Compared to the other figures, I don't feel the need to show this as the main figure. Each illustration is just barely understandable (for example, MRI and wifi router), which also seems to undermine the need for this figure. I suggest moving this figure to SI after revising each illustration and caption, but I would like to leave it to the editor's discretion since this is an editorial comment.

Version 1:

Reviewer comments:

Reviewer #1

(Remarks to the Author)

My main concern was that the main content of this manuscript is too similar to an earlier publication from the same group, and the results from this manuscript do not bring any significant advances, which I believe are required for publication in Nature Communications. My judgment on this remains unchanged.

However, I do find the idea of constructing OPV modules with an output voltage exceeding 1000 V interesting. The authors have addressed my comments from the previous submission and improved the manuscript by performing additional experiments and providing more data. I have no further comments on this manuscript.

Reviewer #2

(Remarks to the Author)

The authors did a very good job on the revision and response to the reviewers. I would suggest its acceptance.

A short question:

"When structuring P3 and P4, the illumination side of the mini-modules were covered with a black tape". What's the reason of black tape covering? please add explanation in the sentence.

Reviewer #3

(Remarks to the Author)

I'm happy to see that the authors addressed all the comments raised by the reviewer appropriately. I can accept this manuscript as is to Nature Communications.

We thank all reviewers for critically assessing our manuscript and showing us where it can be improved. We have considered all comments and suggestions thoroughly, carried out additional experiments and revised the manuscript accordingly. We also added another author who was active in developing the laser process and who was missing in the first submission. Please find our point-by-point answers below.

REVIEWER COMMENTS

Reviewer #1 (Remarks to the Author):

This manuscript by Jiang et al. demonstrates a strategy for constructing ultra-high voltage OPV modules, with laser patterning as its core technique. While achieving such a high output voltage using OPV technology is indeed interesting, the manuscript appears to be heavily technical, lacking in scientific depth. Furthermore, concerns arise regarding the similarity of its core content to an ACS Energy Letters article previously published by the same research group (ACS Energy Lett. 9, 908–910 (2024)), with the primary distinction being an increased number of sub-cells per unit area, resulting in a higher open-circuit voltage. Despite more extensive characterization of the HV module and a proof-of-concept application, the innovative aspects of this work do not surpass those of the previous publication, casting doubt on its suitability for publication in Nature Communications. In my opinion, Scientific Reports may be a more appropriate journal for this research.

Response: We thank the reviewer for this comment. Please note that the paper we published before (ACS Energy Letters) is “only” a highlight paper. We agree that only increasing Voc from 1000 V to 5000 V is not so impressive compared to the new idea itself. However, we would like to point out that our manuscript does not only contain technical aspects but also detailed scientific investigations (e.g. reviewer #3 writes “...analysis for the operation is well discussed”). To the best of our knowledge, previous studies reported in literature about high-voltage photovoltaic modules did not consider the geometric fill factor or how to optimize it and showed results rather without loss analysis. In contrast, the detailed characterization and analysis in our manuscript go well beyond what was previously published, e.g., the quantitative analysis of how shunts impact the device performance as derived from microscopic images, electroluminescence imaging and device simulations as well as the introduction of the break-down index as part of the in-depth reliability testing.

As a matter of fact, the demonstration of powering a DEA suction cup in our manuscript is the first time a single PV mini-module could power a high-voltage (>5000 V) device, without any assist circuitry or battery. However, also here we would like to point out that we do not just show the result of this application but provide a detailed quantitative analysis of the different voltages, the energy flow etc., both in steady-state and transient. This shows that our manuscript contains not only technical data but provides a detailed scientific analysis of the underlying working principles.

We hope that with these explanations the reviewer can reconsider the suitability of our manuscript for publication in Nature Communications.

Recommendations for improvement:

1. It is recommended that the authors provide the EQE spectra of the devices, the LED lamp spectrum utilized for determining PCE, and a detailed explanation of the methodology employed for calculating PCE.

Response: We thank the reviewer for bringing this up. The LED lamp irradiance spectrum and EQE spectra at 100 klux are added as **Fig.2 a,b**. And corresponding text was added:

“ IV measurements were performed with a warm white LED lamp. The irradiance spectrum of the LED lamp is shown in Fig. 2a, with commonly used AM 1.5G as comparison. Fig. 2b shows the external quantum efficiency (EQE) of the solar cells based PV-X plus and PM6:-GS-ISO.”

The PCE was determined in a usual, straightforward manner and we have added the following text to the Methods section:

“With the measured IV curves, the PCE of the HV-OPMs under LED illumination were calculated as:

$$PCE = \frac{I_{mpp} \times V_{mpp}}{5024 \times A_{sub-cell} \times GFF \times P_{in}} \times 100\%$$

where $A_{sub-cell} = 1.2 \text{ mm} \times 0.25 \text{ mm}$, is the full area of a sub-cell, P_{in} the light intensity under the respective illuminance (Supplementary Table 1), I_{mpp} and V_{mpp} are current and voltage at maximum power point, respectively.”

2. The authors should report the maximum output power of the module, as this is a crucial performance indicator.

Response: Please note that the maximum output power of the modules was already included in the Supporting Information as Supplementary Table 1. To emphasize it more for the reader we have added a plot of illuminance dependent Pmax in Fig. 2e.

3. While I acknowledge that shunting is less problematic in HV-devices with a large number of sub-cells, efforts to mitigate shunting and further enhance the performance of the HV-devices would strengthen the work.

Response: We thank the reviewer for this comment and note that also Reviewer #2 made a similar suggestion. In fact, we have made a lot of efforts to mitigate the shunts in both single solar cells and mini-modules. For instance, we used ZnO as ETL, and inserted a thin Au (5 nm) layer between Ag and the HTL layers due to its higher work function. In addition, we carried out more experiments, increasing the layer thicknesses of ZnO and absorber blend, respectively. However, as shown in Figure R1 (it appears again below in the answers to reviewer #2), this approach was not successful as especially thicker absorber blend layers decreased the fill factor, especially at higher intensities (100 klux). Being aware of the importance of shunt mitigation we would nevertheless like to emphasize that the performance we achieved thus far with the high voltage mini-modules down to an illuminance of 1 klux is remarkable.

Fig. R1: Fill factor as a function of illuminance with different spin coating speeds of **a**, PM6:GS-ISO and **b**, PV-X plus.

4. Figures 2b and 2e should be revised for clarity. For instance, please ensure that the connection of adjacent positive and negative electrodes in the schematics is depicted more precisely and intuitively.

Response: We thank the reviewer for pointing this out. The Figure 2b (Fig. 1b in revised manuscript) was revised by adding a zoomed region to show the alignment of the laser lines P1-4 clearly in a microscopic image. Further, Figure 2e was revised by increasing contrast and adding a new sub-plot to show the interconnection between two adjacent rows. The adjacent positive and negative electrodes are now clearly shown by a cross-section, with current flow direction indicated by arrows. Furthermore, we modified the illustration of the increasing potential along two adjacent rows. Corresponding text was modified:

“The current flow direction is illustrated by red arrows in the cross section sub-plot that shows the alignment of P1-P4 clearly.”

5. The authors have provided stability test results for HV-devices under 50k lux conditions. I suggest that, in addition to Voc, data on the degradation of Jsc, FF, and PCE over time be included to provide a more comprehensive evaluation of device stability.

Response: We thank the reviewer for the suggestion. We only showed the Voc data because this is the most important figure of merit in this manuscript. Now, we added the Jsc, FF, and PCE aging data as Figure 4d-f in the revised manuscript. It can be seen that in the first few hours the PCEs are slightly higher than the initial values, mainly because of an increased FF. We attribute this to a modest shunt burning effect occurring during the measurements (i.e., by driving current through the device at voltages beyond Voc). Corresponding text was added:

“ The similar trend was also observed for ISC, FF and PCE. It worth noting that in the beginning several hours, the FF and thus PCE were increased compared to initial values. We attribute to the shunts burning effect during measurement.”

Reviewer #2 (Remarks to the Author):

The paper reports the fabrication and application of very high-voltage organic photovoltaic mini-modules under warm white LEDs. High voltages of 3970 V and 5534 V with two different active layers were achieved on about 15 cm² area. The characterization of dark lock-in thermography and electroluminescence imaging are clear for the understanding of shunt induced voltage losses. The demonstrator for driving a dielectric elastomer actuator is inspiring. I believe this is a piece of very quality work. The demonstration of the high voltage minimodules and its unique application are intriguing. I would recommend its publication in Nat. Communications. Minor comments:

Have the authors tried thicker active layers? Increasing the thickness of the active layers would reduce the possibility of the shunt devices.

Response: We thank the reviewer for these comments.

Yes, we tried thicker active layers and the results are shown in Fig. R1 (at the end of this answer). In principle, it's true that the shunt-proof ability increases with increasing thickness of the active layer. However, it can be seen that with the increase of the active layer thickness, the FF was decreased but not increased, especially for higher degrees of illumination. We attribute this to the transport resistance within the photoactive layer due to the limited charge carrier mobilities. As a result, the transport resistance of the OPV active layer can be the main contribution to the overall series resistance of the device, being larger than the series resistance of the ITO electrode (we have work published on this matter, e.g. DOI: 10.1109/JPHOTOV.2013.2288527; DOI: 10.1063/1.4862960). When substantially increasing the thickness of the active layer, the increasing transport resistance of the active layer can lead to a reduced FF, even under relatively low illumination. In addition to this, there is always a certain degree of shunting as (partial) shunting can also occur due to pinholes in the ETL and/or HTL layers. So even if thicker photoactive layers might reduce overall shunting problems, they also add to a decrease in FF.

Fig. R1: Fill factor as a function of illuminance with different spin coating speeds of **a**, PM6:GS-ISO and **b**, PV-X plus.

Reviewer #3 (Remarks to the Author):

I read this paper with fun. This paper reports Organic photovoltaic mini-module providing more than 5000 volts. They patterns P1-P4 lines using laser processes, and fabricated modules having impressively high Voc and PCE under 100 klux white LED. Finally, they show a demonstration of operating DEA actuator powered by only the OPV module under LED light illumination. The concept is interesting, the fabrication to achieve high voltage using OPVs is well designed, analysis for the operation is well discussed. However, the reviewer suggests revisions before publication in Nature Communications.

1. The current description is reasonable if this paper is submitted to energy related journals since fabrication of high-voltage OPV module is challenging, but the motivation of such high-power output should be clarified more carefully to be published in Nature Communications. Although the authors mention “For many devices, both batteries and voltage amplifiers are bulky”; however, the OPV module used a glass substrate, which should be “bulky” also. Similarly, why do we need the DEA suction cup generating a suction pressure powered only by one HV-OPM without any need for batteries and electronic? What is the major difference between bulky batteries/electronic and rigid OPV module? Actually, there is a report of OPV-driven DEA actuator with support of circuits and batteries, being integrated to make a soft robot system (10.1038/s41598-024-60899-6). Is the procedure applicable to flexible substrates as the authors mentions “the possibility of fabricating mechanically flexible devices”? The relevant discussion should be added to make the justification for using these high voltage OPV modules clearer. Better way is to show the feasibility of these procedure using flexible substrates.

Response: We acknowledge that the glass substrate looks also “bulky” compared to flexible substrates. However, the use of batteries does not only mean bulky, but also brings a limited working duration with it. As mentioned in the manuscript, to ensure the device having high portability/mobility, people usually employ a very small battery. Then the battery does not look so “bulky”, but the working duration is limited to a short time because of low energy capacity. Thus, the major difference between bulky batteries/electronic and rigid OPV modules is that the OPV modules are energy harvesters while the batteries are energy storage devices. The OPV module can work continuously without the problem of a limited energy capacity. In fact, this is the energy autonomy of the DEAs that we addressed in the title of the manuscript. Powering high-voltage devices such as DEAs with only an OPV module constitutes thus a benchmark for real energy autonomy. We also noticed the mentioned paper of powering DEA with OPVs including support of circuits and batteries after initial submission. On the other hand, we also noticed another similar report for sustained flight of an ultralight micro aerial vehicle published in Nature recently (doi.org/10.1038/s41586-024-07609-4). Both reports expanded the applications of PV in the high-voltage field, thus we now cite them in the application part of the revised manuscript as:

“Up to date, few efforts have been made to apply solar cells for high-voltage devices, including DEAs and ultralight micro aerial vehicle^{47,48}. However, in these reports the high-voltage was still generated by high-voltage convertors.”

We would like to point out that the fabrication process of the high voltage organic solar mini-modules is compatible with flexible substrates. In our group we have achieved efficiencies up to 12.5% (under “1 sun”) for 1 cm² active area single cells with the PV-X plus absorber on flexible substrates (ITO on PET). On glass, we typically reach values around 14-14.5% under the same conditions. However, a few engineering challenges have yet to be overcome to realize this also for

the high-voltage mini-modules. In fact, we have tried to build high-voltage mini-modules on flexible substrates (shown in Fig. R2). Unfortunately, all the flexible modules perform poorly because of severe shunts. We attribute this to changing of the laser (the old laser equipment is not available anymore), substrate expansion during thermal annealing, etc. Please note that the laser equipment is not part of our department and the colleagues are usually very busy with other projects. Therefore, it was not possible for us to overcome the mentioned problems within a reasonable time span. In the near future we will get our own laser and then we can identify the best parameters, so future work will definitely be directed towards flexible mini-modules.

We have added the following text at the end of the conclusion section:

“To realize true applications of HV-OPMs, several challenges have to be overcome, such as processing on flexible or even ultra-flexible substrates, miniaturizing control circuit, etc. This is however beyond the scope of this study and will be addressed in future work.”

Fig R2: Photograph of a flexible HV-OPM based on PM6:GS-ISO

2. There are some recent papers related to OPV modules using laser patterning, which are recommended to be referred in this paper. Eg. doi: [10.1016/j.joule.2024.02.016](https://doi.org/10.1016/j.joule.2024.02.016), doi: [10.1038/s41560-022-00997-9](https://doi.org/10.1038/s41560-022-00997-9). The strength of laser patterning can be discussed with referring other patterning strategies such as doi: [10.1016/j.joule.2022.06.015](https://doi.org/10.1016/j.joule.2022.06.015).

Response: We thank the reviewer for pointing this out. We have cited them and a short discussion about the strength of laser patterning was included in the revised manuscript according to the suggestion of the reviewer in the section of “High-voltage organic photovoltaic mini-module design and fabrication” as follows:

“In previous reports, silicon based HV-OPMs fabrication was enabled by a peel-off patterning technique, which is time consuming and complex. Moreover, this method is generally not suitable for OPVs, since the solvent used for photoresist could damage the organic layers. Although multilevel peel-off patterning was already successfully developed for OPVs, the complex process makes it still not be widely adapted²⁰. On the contrary, direct laser patterning is powerful for micron-scale structuring, which is widely used for thin film based high efficiency PV module fabrication, due to its low cost, fast processing and minimized geometrical loss²¹⁻²³.”

3. Since Nature Communications have wider readership, detailed information should not be omitted. P1-P4 are understandable words for solar cell field researchers but are not at all for others. They used pico-second laser to make the patterns according to the reference 19, but this is not specified in this paper (Additionally, reference 19 does not show product details of the laser).

Response: We thank the reviewer for the suggestion. The description of P1-P4 was added in the revised manuscript as *“For two adjacent sub-cells in one row, the bottom ITO and top Au/Ag electrodes were separated by P1 and P3 lines, respectively, and interconnected by a P2 line. Two adjacent rows were separated by a P4 line.”* in the section of “High-voltage organic photovoltaic mini-module design and fabrication”. Further, the product details of the laser were added in the method section as following:

“The laser system used was a UV ultra-short pulse system from Lumemntum in conjunction with a fixed optics and an axis system from Aerotech „Pro Serie“ to move the sample.”

4. To me, 19% under 100 klux is too high since this is almost similar to 1-Sun condition, which maybe related to spectrum of LED light. The irradiance results of LED light with 100 klux (and others also) should be added to verify the calculation of efficiency. Additionally, I suggest showing the PCE under AM1.5G illumination conditions as well as LED light conditions to avoid misunderstanding.

Response: Indeed, the high efficiency under 100 klux is (very) strongly related to the spectrum of the LED light which is much narrower compared to sunlight, thereby significantly reducing thermalization losses. As a result, the PCE is higher. To make it clear, we present the LED spectrum together with the AM 1.5G spectrum in Fig. 2a of the revised manuscript and show the estimated efficiency under AM 1.5G as 10.2% for PV-X plus based modules, with detailed calculation in Supplementary Note 1 (revised version).

5. “As shown in Fig. 5e, the BRI has a negative value for both PM6:GS-ISO and PV-X plus HV-OPMs for any value of Nsha, which means that there is no breakdown risk at all for illuminances of 100 klux and below” (Page 13); is there breakdown risk under AM1.5G condition?

Response: The answer is no. Even there is more power dissipated in the shaded sub-cells under AM 1.5G illumination, the BRI is still far below 0, which means no risk of breakdown. The corresponding analysis is shown in Supplementary Fig. 11-14. The values for AM 1.5G have been added as dashed line to Fig. 4h.

6. Why is Fig.4f much more blur than Fig.4d? Is this related to the performance of the module?

Response: We thank the reviewer for this comment. In fact, we have tried our best to make the Figure 4f as clear as the Figure 4d but failed and we do not know the origin.

7. The purpose of Figure 1 is not clear. Compared to the other figures, I don't feel the need to show this as the main figure. Each illustration is just barely understandable (for example, MRI and wifi router), which also seems to undermine the need for this figure. I suggest moving this figure to SI

after revising each illustration and caption, but I would like to leave it to the editor's discretion since this is an editorial comment.

Response: We thank the reviewer for this suggestion. Our initial motivation to add Figure 1 was that we wanted to emphasise the lack of low-power high-voltage sources for emerging devices based on dielectric elastomer actuators, electroaerodynamic thrusters, etc. The conventional applications and power supply methods were just shown for comparison. After a careful evaluation, we agree with the reviewer that it would be better to delete it. Correspondingly, the following sentences were added to the main text and Figure 1 was removed in the revised manuscript:

“ Usually, low-power low-voltage portable devices are powered by batteries, while unportable/immobile devices/equipment are powered by an electrical (power) grid. In these conventional scenarios, the high-voltage devices/equipment are usually high-power too. However, there are more and more high-voltage but low-power devices developed in recent years...”

REVIEWERS' COMMENTS

Reviewer #1 (Remarks to the Author):

My main concern was that the main content of this manuscript is too similar to an earlier publication from the same group, and the results from this manuscript do not bring any significant advances, which I believe are required for publication in Nature Communications. My judgment on this remains unchanged.

However, I do find the idea of constructing OPV modules with an output voltage exceeding 1000 V interesting. The authors have addressed my comments from the previous submission and improved the manuscript by performing additional experiments and providing more data. I have no further comments on this manuscript.

Answer: We can only repeat that this manuscript differs significantly from our previous publication. In the latter, we only presented the proof of principle that an organic photovoltaic mini-module could work and reach a voltage of more than 1000 V. In the work submitted to Nature Communications, we go far beyond this. A detailed analysis of the working principles and the factors limiting device performance was carried out and delivered substantial scientific insight into the specific nature of these high voltage mini-modules. Apart, we could further boost the voltage and demonstrate a use-case by powering the dielectric elastomer micro-actuator based suction cup.

Reviewer #2 (Remarks to the Author):

The authors did a very good job on the revision and response to the reviewers. I would suggest its acceptance.

A short question:

"When structuring P3 and P4, the illumination side of the mini-modules were covered with a black tape". What's the reason of black tape covering? please add explanation in the sentence.

Answer: we thank the reviewer for bringing up that point. The reason is that we have to avoid an electrical breakdown as an already very high voltage is generated by the ambient light during the laser processing as the module can only be encapsulated after all laser processing is finished (note that the encapsulation protects the module from suffering an electrical breakdown). We have added this information to the manuscript.

Reviewer #3 (Remarks to the Author):

I'm happy to see that the authors addressed all the comments raised by the reviewer appropriately. I can accept this manuscript as is to Nature Communications.

Answer: We thank the reviewer for the assessment of our manuscript and for this positive answer.